

# Real-time plotting and evaluation of the data quality control from the CSIR- NGRI Magnetic observatories

Vengala. Pavan Kumar, Nelapatla. Phani Chandrasekhar and Potharaju. Sai Vijay Kumar

CSIR-National Geophysical Research Institute, Hyderabad, India

Corresponding author: phaninelapatla@gmail.com

**Abstract**

Earth's magnetic field, a dynamic shield influenced by internal and external forces, holds critical insights into space weather forecasting and the planet's core dynamics. The Choutuppal (CPL) and Hyderabad (HYB) magnetic observatories in India are pioneering this field by delivering high-resolution geomagnetic data to INTERMAGNET with unprecedented speed and precision. Utilizing a novel, low-cost protocol, CPL transmits 1s resolution data and HYB provides 1 min data, both achieving a latency of less than 300s making them among the first observatories worldwide to accomplish this feat. This rapid data transmission enhances global collaboration in space weather prediction, safeguarding critical infrastructure like satellites and power grids from solar storms.

To further elevate data utility, we developed a Python based software for real-time visualization and quality control at both observatories. This tool generates plots, performs initial quality checks, and computes first differences at 1s and 1min intervals, with a latency under 300s. By enabling daily evaluation of data quality, the software facilitates the identification of anomalies and noise, supporting the preparation of quasi-definitive data essential for geomagnetic research. Our Python server and web applications are designed with the future in mind, integrating artificial intelligence (AI) and machine learning (ML) capabilities. These advancements at CPL and HYB are set to transform the processing, forecasting, and visualization of geomagnetic data. By improving both the accuracy and accessibility of this data, we aim to revolutionize geomagnetic research, making it more precise, accessible, and actionable.

## 1. Introduction

Geomagnetic observatories are fixed locations on the Earth's surface that continuously monitor the geomagnetic field. The continuous time series data from these observatories reflect various physical processes associated with solar and Earth activities. The reliability and accuracy of the data are crucial for both scientific research and commercial applications [Matzka et al. 2010; Zhang et al. 2016]. Each observatory is equipped with at least two tri-axial fluxgates to record variations along with Overhauser magnetometers. Additionally, it has one fluxgate theodolite and Overhauser / Proton precession magnetometer for absolute measurements.

The Hyderabad Magnetic Observatory (HYB) of CSIR-NGRI has maintained 60 years of uninterrupted and stable recordings of magnetic variations. In 2009, with upgraded instruments, HYB became an INTERMAGNET observatory. Due to rapid urbanization and the introduction of the Hyderabad Metro Rail project nearby, it became essential to establish an alternate observatory to ensure the continuity of the geomagnetic data series. The campus of the former Choutuppal (CPL) Geo-electric Observatory provided an ideal location for recording magnetic measurements at one-second intervals. Preliminary 1 min observations began in 2012 and 1s data recording was initiated in 2015, and the ongoing data collection has





led to CPL being recognized as an INTERMAGNET observatory in 2019. The high-quality 1s
definitive data from CPL is currently being submitted to INTERMAGNET, making it one of
the first observatories in this region to achieve this status [Arora et al. 2016]. The HYB and
CPL observations have made significant contributions to global data, alongside other
observatories worldwide, for the main model of the Earth's magnetic field. These observations
have also supported various studies of low-latitude magnetic phenomena [Dwivedi and
Chandrasekhar, 2024] and regional induction anomalies [Edara and Arora, 2023].
For any observatory, the quality of the data is very important to achieve and maintain
INTERMAGNET status [Clarke et al. 2013]. Many scholars have developed various quality
control methods for geomagnetic data [Curto and Marsal 2007 and references therein]. Clarke
et al. [2013] developed an automated data processing software that integrates daily extrapolated
baseline values of H, D, and Z, derived from baseline functions, with H, D, and Z variometers
data. 1min data are delivered to the Edinburgh INTERMAGNET Geomagnetic Information
Node (GIN) in near real-time and on the following day. After implementing a few procedures,
the quality data are prepared and delivered to GIN by running the data processing software in
manual mode on the next working day. Chandrasekhar et al. [2017] discussed the challenges
involved in measuring 1s variations in the geomagnetic field to meet the standards set by
INTERMAGNET for quality and data transmission at observatories over extended periods.
They also provided a detailed account of the progressive steps that led to the successful
establishment of these measurements at the CPL observatory. Khumotov et al. [2017]
developed a new method for the noise identification in the identification in the data at
observatories of IKIR FEB RAS (Russia) and CSIR-NGRI (India). They also presented a
review of commonly used methods for noise identification in practical situations, highlighting
the potential for reducing the impact of noise on data through various examples. He et al. [2022]
proposed a method that combines genetic algorithms and linear regression to evaluate
geomagnetic data quality. Their approach considers factors such as observational data, attitude
angle, scale factor, long-term drift, and temperature. They highlight that agreement among
geomagnetic vector observations is crucial for assessing data quality and utilize Bland-Altman
plots, applying a 95% confidence interval to evaluate this agreement quantitatively and
qualitatively. Lingala et al. [2022] discussed the observed noticeable differences in the noise
levels present in vector and scalar variation data, due to the vehicular noise observed before
and during the COVID lockdown period and also discussed the details of increased data quality
in the absence of traffic-generated noise sources. da Silva et al. [2023] developed the Magnetic
Observatories and Stations Filtering Tool (MOSFiT), a Python package designed to visualize
and filter data from magnetic observatories and magnetometer stations. This tool can also be
utilized for quality control of geomagnetic observatory data, similar to the methods
implemented by the British Geological Survey as described by Macmillan and Olsen [2013].
Several studies have discussed the quality of geomagnetic observatory data and improved
protocols for addressing noise in the data [Zhang et al. 2024 and references therein].
Apart from the data quality, the other important aspect of any Geomagnetic observatory is
remote site data transfer, which is crucial for various applications, including environmental
monitoring, scientific research, and etc. All the INTERMAGNET Observatories data world-
wide is collected by the Geomagnetic Information Nodes (GINs) serve as collection points for
real-time data and are connected to the INTERMAGNET Observatories through satellite,
computer, and telephone networks. GINs operate in five different countries: the UK, USA,





Japan, Canada, and France. They utilize four satellites: GOES-E, GOES-W, METEOSAT, and
GMS to receive real-time data from INTERMAGNET Observatories worldwide
(https://intermagnet.org/gins.html)
Numerous observatories around the globe transfer 1-minute to 1-second data in real time to
various GINs using different technologies. These technologies include satellite
communication, ISDN telephone links, FTP, VPN router servers, in-house built NDL HSS,
MQTT, and various third-party software and tools [Torta et al. 2009; Clarke et al. 2013;
Chulliat and Chambodut, 2014; Thomson, 2014; Gvishiani et al. 2016; Reda and Neska, 2016;
Zhang et al. 2016]
Recently, Potharaju and Nelaptla [2023] addressed the challenges of data transmission for both
1s and 1 min intervals. They detailed the step-by-step development process, algorithm creation,
function libraries, and the implementation of real-time data transmission from a remote
observatory. Using the Python programming language, they developed an algorithm to
automate the transmission of high-resolution real-time magnetic data from CPL and HYB to
Edinburgh GIN, all while relying on minimal internet connectivity. The automation system
securely transfers data in an encrypted manner using SSH keys, while also saving the same
dataset on a local server at CSIR-NGRI. Data from both observatories is sent to GIN in real-
time within a timeframe of less than 300 seconds. After successfully transmitting 1-minute
geomagnetic data in real-time from the CPL and HYB observatories to Edinburgh GIN, the
transmission of CPL's 1-second real-time data have also commenced. This achievement marks
CPL as one of the first Indian observatories to send 1-second real-time data to GIN.
In this paper, we outline the processes involved in upgrading the Python tool package that
facilitates real-time data quality control checks at the CSIR-NGRI magnetic observatories,
HYB and CPL, for both 1-second and 1-minute data intervals. Additionally, we discuss the
various options for installing and implementing our package, ensuring it integrates smoothly
with the available resources for real-time data transmission and quality checks.

## 115  2. Real-time data transfer to INTERMAGNET GIN

The real-time geomagnetic data, both vector and scalar, is initially collected by the MAGREC-
4B logger. This data is then transmitted to a local machine running the CentOS operating
system, which is deployed at the observatory (CPL/HYB). A secure communication is
established via SSH (Secure Shell) using key-based authentication. Initially, an SSH key pair
is generated, and to enhance security, this key is changed every two weeks to prevent
unauthorized access.
Once the data is received on the CentOS machine, it is processed and prepared for transfer to
the centralized server located at the NGRI-HYB Observatory. The secure transfer to the
observatory server is conducted using the same SSH protocol, ensuring a robust and encrypted
handshake for optimal data integrity and confidentiality. Upon arrival at the NGRI-HYB
Observatory server, the collected data is methodically organized based on its temporal
granularity. The data is categorized and stored in specific directories, namely "Minute Data"
and "Second Data," facilitating organized data management and easy access for analysis.
The segregated data from the HYB server is then prepared for transmission to the
INTERMAGNET Geomagnetic Information Node (GIN) located in Edinburgh, UK. This
transmission process is automated using Python scripts and daemon processes that run in the
background. These scripts are designed to execute every 300 seconds (5 minutes), ensuring



timely and regular data updates without duplication. The Python code handles data packaging,
error checking, and retransmission logic to ensure reliable data delivery.
Throughout the entire process, various security measures are implemented, including regular
updates to SSH keys and continuous monitoring of data transfer processes. Logs are maintained
to track data transfer activities and to quickly identify and rectify any anomalies or issues. By
employing this comprehensive workflow, the system guarantees secure, efficient, and reliable
data transfer from the MAGREC-4B logger to the INTERMAGNET GIN, thereby facilitating
continuous monitoring and analysis of geomagnetic data [Potharaju and Nelaptla, 2023].
**3. Customization of the PHP server for real-time data visualization**
PHP (Hypertext Preprocessor) (*https://www.php.net/*) is a widely-used server-side scripting
language primarily designed for web development. It is embedded within HTML and executed
on the server, generating dynamic content that can be displayed on web browsers. PHP offers
flexibility, simplicity, and compatibility with various databases, making it a popular choice for
developing interactive and data-driven web applications. Real-time data visualization is
essential in various domains, including geomagnetic observatories, seismic monitoring and IoT
applications. PHP, in combination with Plotly (a JavaScript-based visualization library)
(*https://plotly.com/javascript/*), facilitates the rendering of real-time plots by handling server-
side data processing and sending the results to the client.
Since data from both observatories is available in real-time, we have developed a PHP server
that simultaneously plots the data from both locations. The screen refreshes every 300s to
display the updated trends for each component and continues till the end of the day. This server
is designed to store initially a weekday's data at one-minute sampling rate, as illustrated in
Figure 1.

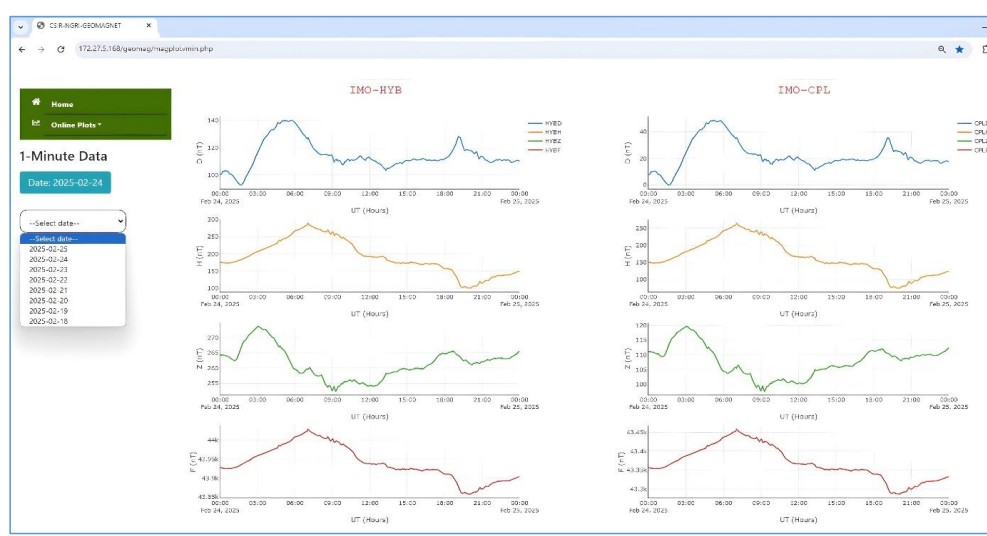

Figure 1. Real-time plotting of Vector and Scalar data at 1 min sampling interval from the HYB
(left) and CPL (right) observatories of CSIR-NGRI




**4. Upgrading the PHP server to a Python server**

Real-time data visualization is crucial in various fields, including geomagnetism, seismology, seismic, and many other Geophysical applications, where continuous monitoring and dynamic plotting are essential. Traditionally, PHP has been a popular choice for web-based data visualization due to its simplicity and widespread usage. However, with the increasing demand for scalability, performance, and flexibility, Python Django (*https://www.djangoproject.com/*) has emerged as a more powerful alternative.

Upgrading from PHP to Django for real-time plotting provides significant advantages in terms of performance, scalability, and maintainability. Django's powerful framework, combined with Python's rich ecosystem, allows efficient handling of large datasets, real-time updates, and seamless integration with machine learning models. Although the migration process involves challenges such as database compatibility and code refactoring, the long-term benefits in flexibility, performance, and extensibility make Django a superior choice for real-time plotting applications.

Django is a high-level Python web framework that promotes rapid development and clean design. Python's libraries (e.g., Pandas, NumPy) enable complex data analysis and enhances the efficient backend processing. Django channels and WebSockets provide low-latency, real-time data streaming. It's ORM (Object-Relational Mapping) simplifies database interactions. Django provides a modular architecture, making the application easier to scale and maintain.

Django, combined with **Bokeh (***https://bokeh.org/*** )**, a powerful visualization library, offers an efficient solution for rendering real-time plots. Bokeh's interactive plotting capabilities, when integrated with Django's backend, enable dynamic and responsive data visualizations. Bokeh is a Python library for creating interactive and real-time visualizations, it generates JavaScript-powered visualizations directly in the browser. It supports streaming data sources for dynamic, live-updating plots. It provides interactive tools like panning, zooming, and hovering functionalities for better data exploration.

With the increasing availability of geomagnetic data from observatories and satellites, Artificial Intelligence (AI) and Machine Learning (ML) techniques are transforming the field by enabling: a) Real-time monitoring and forecasting of geomagnetic events, b) Anomaly detection for space weather and magnetic storms, c) Data-driven insights for understanding magnetic field variations, d) Predictive models for geomagnetic hazards.

To achieve above mentioned applications, Python offers a robust ecosystem of AI/ML libraries that can be directly integrated into Django-based geomagnetic applications. Django serves as the API layer or backend framework for delivering AI/ML model predictions as web services. Django + Python + AI/ML provides a future-proof, scalable, and efficient framework for geomagnetic data processing, visualization, and prediction.

**5. First difference tool for real-time data quality checks**

The "first difference" (FD) is a key analytical tool used in time series analysis. It refers to the difference in values between consecutive observations in a dataset. This method transforms a time series dataset to make it stationary, which helps in identifying patterns, trends, and other dynamic aspects over time more easily of a signal. It is particularly useful for analysing



geomagnetic time series data and understanding its evolution. If the difference between two
consecutive time periods of a signal is abnormal, it may indicate the presence of noise in the
data, often caused by anthropogenic / environmental factors.
We have developed a real-time FD tool in Python that can calculate data for each component
of both observatories at 1s and 1 min intervals, allowing for a quick and hassle-free assessment
of data quality. This computation refreshes every 300s and displays the differences. Here is an
example that illustrates the 1-minute plot (Figure 2) and the 1-second plot (Figure 3) for
February 7, 2025, for the HYB (left panel) and CPL (right panel) observatories across each
component. We have upgraded the server to include several months of data, enabling users to
access the desired day instantly as needed. One example is illustrated in Figures 2 and 3.

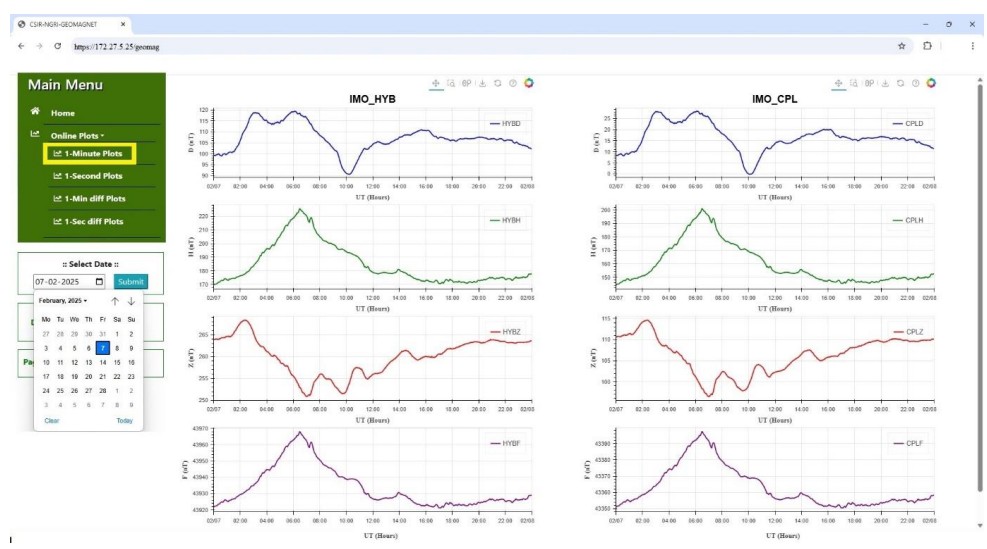

Figure 2: Real-time plotting of Vector and Scalar data at 1 min sampling interval from the HYB
(left) and CPL (right) observatories of CSIR-NGRI, with the updated services in the server.



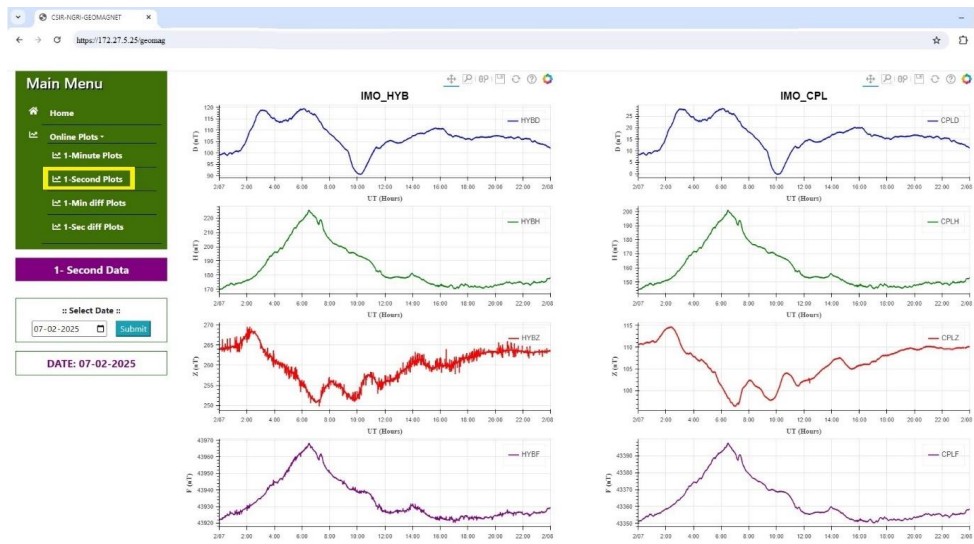

Figure 3: Real-time plotting of Vector and Scalar data at 1s sampling interval from the HYB
(left) and CPL (right) observatories of CSIR-NGRI.
Upon reviewing Figures 2 and 3, we observe that the noon hours of the day began with a sudden
storm commencement (SSC). This term refers to an abrupt increase or decrease in the
northward component of the geomagnetic field and indicates the onset of a geomagnetic storm.
The SSC event is noted around 07:14 UT at both observatories. In comparison to 1s, 1 min data
exhibits more noise in the Z and F components at HYB than at CPL (Figures 2 and 3).

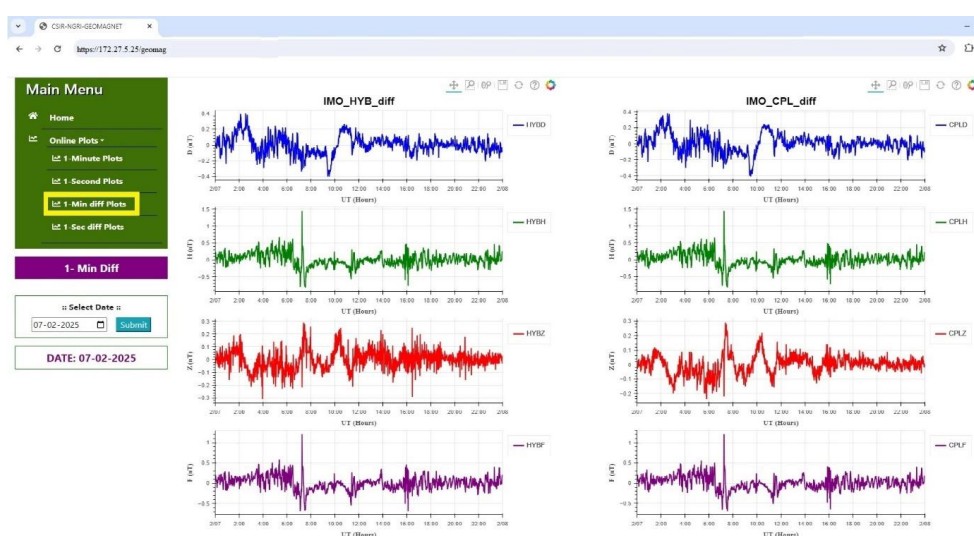

Figure 4: Real-time plotting of the first differences of vector and scalar data, sampled every
minute from the HYB (left) and CPL (right) observatories



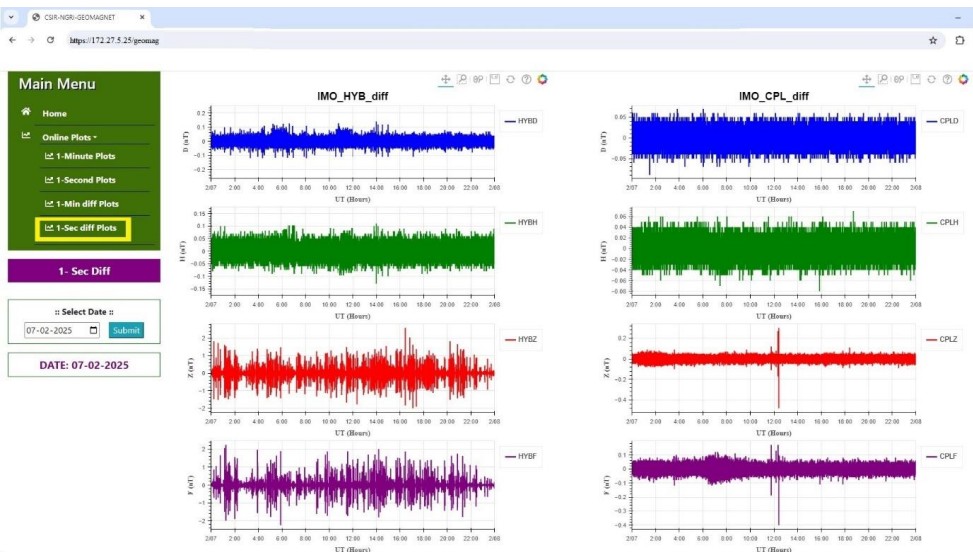

Figure 5: Real-time plotting of the first differences of vector and scalar data, sampled every second from the HYB (left) and CPL (right) observatories

Figures 4 and 5 present the calculated FD of each component at both the HYB and CPL observatories, measured in 1 min and 1s intervals. From these figures, it is apparent that small amplitude spikes were observed at the CPL observatory, particularly in the Z and F components around (12:21:07UT, and 12:24:57UT) during that day. The amplitude of these spikes at CPL are about: 0.4 nT and 1.5 nT. Further, the amplitude of the FD's of all the components are in the range of ±0.5nT in D, ± 1.5nT in H, ± 0.3 in Z and ± 1.5nT in F for 1 min data and ± 0.1 nT in D, ± 0.1 nT in H, ± 0.5 nT in Z at CPL, ± 2 nT in Z at HYB, ± 0.5 nT in F at CPL and ± 2 in F at HYB for 1s data. The next step is treating the spikes by evaluating the source behind the signal. After checking the logs, the spikes recorded at CPL are result from human intervention related to data collection from the flashcard of the spare fluxgate magnetometer deployed in the sensor hut. Hence the observed spikes at the components (D, H, Z and F are removed from the data set. In contrast, a greater number of spikes were recorded in the Z and F components at the HYB observatory on 7[th] February 2025. It is important to note that the spikes recorded in the Z and F components at HYB are attributed to vehicular traffic and metro rail operations, as discussed by Lingala et al. [2022]. These activities occur regardless of the geomagnetic conditions. Therefore, this data requires a treatment of noise removal before it can be submitted as quasi-definitive data. After treating the noise in the data at both the observatories, the final data for HYB and CPL for 1 min are shown in Figure 6.



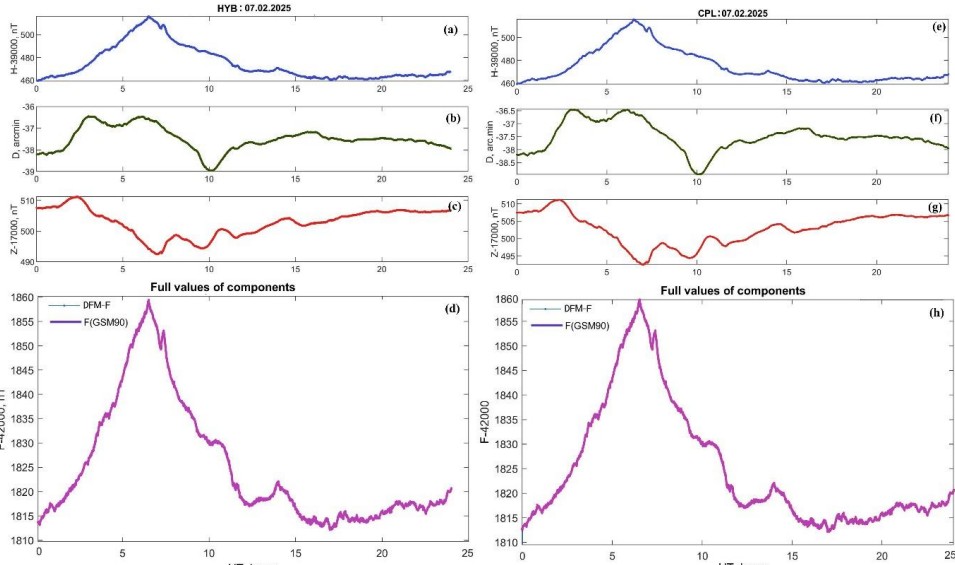

Figure 6: 1 min vector and scalar data from the HYB (left) and CPL (right) observatories after removing spikes identified through real-time first differences.

The plotting page also offers the following tools:

(a) **PAN**: This tool allows you to move the view of an image or document around on the screen while maintaining the same zoom level. Essentially, it enables you to "slide" the image to view different parts without zooming in or out.

(b) **Box Zoom**: This zoom method lets you click and drag to create a rectangular area on the screen, specifying which part of the content you want to zoom in on. This tool is particularly useful for identifying noise in the data.

(c) **Wheel Zoom**: This tool lets you zoom in and out using the mouse scroll wheel.
(d) **Reset**: This tool restores the view to the normal plotting window, removing any Box Zoom or Wheel Zoom adjustments.

(e) **Save**: This function allows you to download the current plot as a ".png" image format.

**6. The essence of extending the Python based facility to Remote Sites**

CSIR-NGRI currently operates remote sites at various locations: Himalayas, Andaman & Nicobar Islands, and Ladakh. Extending the existing data transfer facility to these remote locations offers several key benefits:

- **Consistency:** Ensures uniform data collection and transmission across multiple sites, resulting in consistent and reliable datasets.





- **Scalability:** Facilitates easy expansion of the network to include more remote locations without incurring significant additional costs.
- **Resource Optimization:** Maximizes the use of existing infrastructure and resources, reducing the necessity for redundant systems.
- **Enhanced Data Insights:** Provides a broader range of data, leading to more comprehensive analyses and improved decision-making.
- **Cost Efficiency:** By utilizing the same setup, organizations can save on costs associated with deploying and maintaining separate systems for each site.

All remote site recording systems are equipped with maintenance-free batteries that are regularly charged by solar panels. We have identified several potential systems for installing the developed Python-based package, which is designed to implement low-cost and effective data transfer techniques and tools, along with real-time quality control checks in the next few months.

**A. Techniques:**

**1. *LoRaWAN*:** A low-power, long-range wireless communication technology that is ideal for remote areas with minimal infrastructure. It is cost-effective and suitable for IoT applications.

**2. *Satellite Communication*:** Using Low Earth Orbit (LEO) satellites can provide reliable internet connectivity in remote locations. Companies like Soracom and Sateliot offer affordable satellite IoT solutions.

**3. *Store-and-Forward Systems*:** Data loggers can store data locally and transfer it when a connection is available. This method is beneficial in areas with intermittent connectivity.

**4. *High-Frequency Radio Communication*:** HF radio can be employed for long-distance communication in remote areas, requiring a clear line of sight, and can be a cost-effective solution.

**B. Tools:**

**1. *Airbyte*:** This tool supports incremental data synchronization, making it efficient and resource-effective. It primarily runs on Linux-based operating systems, specifically Ubuntu and Debian. Although it can also be deployed using Docker on Windows and macOS, Linux is the preferred operating system.

**2. *Raspberry Pi with SIM Card*:** This versatile and cost-effective option is equipped with a SIM card to collect and transmit data. It typically runs on Raspberry Pi OS (formerly known as Raspbian), which is based on Debian Linux, but it also supports Ubuntu, Alpine Linux, Arch Linux, and other Linux distributions.

**3. *Omega2 LTE*:** This single-board computer (SBC) provides high-speed cellular connectivity with LTE Cat 4 support, making it ideal for remote data transfer. It operates on OpenWRT, a lightweight Linux-based operating system optimized for networking and IoT applications.

**4. *Libre Computer Board Le Potato*:** While primarily focused on multimedia applications, this board can be adapted for data transfer using SIM card technology. It mainly runs on Ubuntu, Debian, and Armbian (a lightweight Linux distribution designed for ARM-based SBCs).



**5. *Orange Pi 5*:** This board supports multiple Linux-based operating systems, including
Ubuntu, Debian, Armbian, and Android. It offers better performance than the Raspberry Pi at
a similar price point and is compatible with SIM card technology for data transmission.
By selecting any of the above approaches will enhance the capabilities of CSIR-NGRI's remote
operations, ensuring effective data management and communication across challenging
environments
**7. Summary**
Raw magnetic observatory data often contains noise and artifacts that need to be removed or
corrected. Pre-processing steps include the removal of spikes, correction for temperature
effects, time synchronization, and baseline adjustments. Data quality control is essential to
ensure the reliability of the information, involving visual inspections, statistical analyses, and
data flagging.
Many researchers prepare quasi-definitive data on a monthly, weekly, or daily basis by
incorporating these pre-processing steps and submit their findings to INTERMAGNET GIN.
For instance, the IPGP quasi-definitive data method is a monthly process focusing on obtaining
the most accurate results for the recent past [Peltier and Chulliat, 2010]. The BGS method,
while similar, also aims to produce next-day quasi-definitive data using predicted baseline
values [Clarke et al. 2013]. Both methods are valid for meeting the quasi-definitive data
definitions set by INTERMAGNET and offer distinct strengths to benefit various data users.
da Silva et al. [2023] introduced the Python package MOSFiT, designed to work with 1-minute
INTERMAGNET definitive and quasi-definitive data, but it can also be applied to any
geomagnetic observatory or magnetometer data. The CSIR-NGRI geomagnetic observatories
at HYB and CPL participate in and contribute to INTERMAGNET and submitting quasi and
definitive data. Although both observatories are equipped with the same magnetometers, the
CPL observatory operates in a noise-free environment, while HYB does not. CPL is India's
first observatory to provide 1-second data, whereas HYB offers 1-minute real-time data to GIN.
Before processing data to produce quasi-definitive results, it is vital to validate its quality. To
address this need, we have developed a Python-based plotting service tool as the first step in
our data quality control process. This tool not only monitors real-time trends and continuity in
observational data but also includes a dedicated review process to rigorously assess data quality
regularly. This ensures the accurate preparation of quasi-definitive data for the observatories.
Additionally, the tool has indicators that flag data when the FD values exceed specified
thresholds, ensuring accuracy and completeness.
Our Python-based tool can be installed on various client-side devices, including LoRaWAN,
data loggers, Airbyte, Raspberry Pi, Omega2 LTE, Libre Computer Board Le Potato, and
Orange Pi 5. These devices are suitable for deployment in remote locations with limited power
availability. On the server side, the system can be configured to connect to a workstation or
server to receive data in real-time. The establishment of this real-time quality control system
significantly enhances the data quality from both permanent and remote observatory sites,
providing reliable support for related scientific research.
Further, our Python-based server is designed to provide a robust ecosystem of AI/ML libraries
that can be seamlessly integrated into Django-based geomagnetic applications. Django serves
as the API layer and backend framework for delivering AI/ML model predictions as web



services. The combination of Django, Python, and AI/ML creates a future-proof, scalable, and
efficient framework for processing, visualizing, and predicting geomagnetic data.
**8. Conclusions**
High-quality magnetic observatory data is vital for understanding the Earth's magnetic field.
Observatories in exceptional locations provide valuable long-term data, but assessing data
quality requires expertise and can be time-consuming. Producing reliable geomagnetic data is
challenging, especially for institutions with limited staff. Rising operational costs can make it
difficult to secure necessary funding. Despite these hurdles, many scientific publications rely
heavily on high-quality absolute magnetic observatory data.
We have developed a Python-based tool to assist observatory staff in identifying noise in data,
which will help reduce their workload. In the future, we plan to create a scalable framework
for processing, visualizing, and predicting geomagnetic data using Django, Python, and AI/ML
technologies. Additionally, we have established a cost-effective data transfer system that
enables reliable data collection and analysis from remote locations without imposing
significant financial burdens, thereby benefiting organizations with limited budgets.

**Acknowledgements**
The authors thank the Director of CSIR-National Geophysical Research Institute (Hyderabad)
for the support and permission to publish this work (NGRI/Lib/2025/Pub-45). The authors wish
to thank Dr. Kusumita Arora for her constant encouragement and permission to carry out this
research work.

**Author contributions**
VPK (V. Pavan Kumar), NPCS (N. Phani Chandrasekhar) and PSVK. (P. Sai Vijay Kumar)
wrote the manuscript and reviewed the article, NPCS prepared figures.

**Competing interests**
The authors declare no competing interests.

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
