# Peer review of "Real-time plotting and evaluation of the data quality control from the CSIR- NGRI Magnetic observatories"

_EGUsphere, 2025_

## Author Response (AR1)

Reply to RC1 comments

Comment-1: -The authors have developed Python-based tools for real-time visualisation and quality control of one-second and one-minute geomagnetic data. The initial automated data control focuses mainly on analysing "first differences". Implementing the "first difference" method effectively detects and flags anthropogenic disturbances, making it easier to remove these disturbances before publishing Quasi-Definitive data. However the article does not mention the commonly used data control method based on the Fv-Fs difference. It is unclear whether this method is not used at the HYB and CPL observatories. If it is used, it should be mentioned in the manuscript; if not, the authors should explain why it was omitted.

Reply: We thank the reviewer for highlighting the omission of the Fv-Fs difference method in our manuscript and for seeking clarification on its use at the HYB and CPL observatories. We acknowledge the importance of this widely recognized data control technique and confirm that it is actively employed at both observatories to ensure high-quality geomagnetic data. Lines: 397-410.

Comment-2: Lines 13, 106, Intermagnet Manual https://tech-man.intermagnet.org/stable/chapters/submitdata/introduction.html
chapter 6.4.1 says about goals. For near real-time performance: 30s for 1s-data, 60s for 1m-data. About 50% of observatories provide 1m data with a delay <= 5 minutes. The relevant statistics are available at https://imag-data.bgs.ac.uk/GIN_V1/GINStatistics. The claim "achieving a latency of under 300 seconds and being one of the first observatories worldwide" is a bit debatable.

Reply: As suggested, we have gone through the manual and the link providing the statistics on the real-time data transmission. The following are our observations:

Other observatories, such as BEL, HEL, and Hornsund in Poland, achieve a latency of around 5 minutes by utilizing VPN routers and backup servers, indicating the use of robust data transmission infrastructure. In comparison, NRCan, BGS, and ASP observatories employ satellite links and high-performance servers, reflecting their adoption of more advanced communication systems suitable for their operational requirements.

On the other hand, the CPL, HYB, and TTB observatories seem to be better candidates based on the available evidence. These magnetic observatories transmit live data to INTERMAGNET GIN with a latency of 5 minutes or less on a daily basis, using low-cost, low-resource setups that do not require heavy

infrastructure. While 50% of observatories provide 1-minute data with a delay of 5 minutes or less, the time frame varies from a minimum of 2 minutes to a maximum of several thousand minutes. However, detailed information about the infrastructure used for data transmission is lacking. Therefore, it is noted that CPL and HYB utilize lightweight data transfer systems using Python, rsync, and broadband service along with basic desktop computers. Meanwhile, TTB employs a minimal Raspberry Pi setup, which fulfills all necessary requirements for a cost-effective data transmission system.

Comment-3: Lines 89-91, Although these satellites were used in the past, the preferred way to send data to the GINs now is through the internet, and satellite channels are only used as a backup option.

Reply: Included in the text as suggested. Lines: 91-93.

Comment-4: Line 132, "300 seconds (5 minutes)" – (5 minutes) is not necessary

Reply: Modified 5 mins as 300s throughout the manuscript as suggested.

Comment-5: Line 423, It is worth adding the DOI https://doi.org/10.1007/s00024-023-03333-8)

Reply: Included DOI as suggested

Comment-6: Line 432, it is worth adding the DOI (https://doi.org/10.5194/gi-6-329-2017)

Reply: Included DOI as suggested

Comment-7: Line 449, It is worth adding the DOI (https://doi.org/10.4401/ag-4572)

Reply: Included DOI as suggested

Comment-8: Many lines, it would be good to make the data names consistent throughout the manuscript. Now, the one-second data is called both '1s' and '1-second'. The same issue concerns one-minute data.

Reply: Modified the data names throughout the manuscript as suggested.

Comment-9: Lines 33, 46, 52, 95, 96, 97, 339, year should be preceded by a comma

Reply:  Included comma as suggested

Comment-10: Lines 98, 140, publication year should be rather 2022, Nelaptla (typo error)

Reply:  Corrected the typo error as suggested

Comment-11: Line 673, Khumutov (typo error)

Reply:  Corrected the typo error as suggested

Comment-12: Lines 156, 214, 220, 230, 234, 258. All figures are low quality. For instance, the axis labels are difficult to read.

 Reply:  Improved the resolution of the figures and font size increased for axis labels as suggested.

Reply to RC2 comments

Comment-1: -The authors cite a few related services used by other observatories (Khumutov et al., 2017; He et al., 2022; and MOSFiT by da Silva et al., 2023). Another relevant tool to consider is the MagPy package maintained by INTERMAGNET (https://github.com/geomagpy/magpy), which is widely used by observatories and provides similar features, including data plotting and automatic spike detection. A more precise comparison between the proposed system and existing tools (MagPy, MOSFiT, etc.) would strengthen the manuscript and clarify the novelty and advantages of the new system.

Reply: Here's a precise comparison between MagPy, MOSFiT, and real-time first differences in geomagnetism based on their applications, methodologies, and use cases in geomagnetic data analysis:

1. MagPy is a Python library specifically designed for processing and analyzing geomagnetic data. It offers features such as data filtering, visualization, and quality control. MagPy provides tools for baseline correction and noise removal, as well as the ability to compute differences (e.g., between observatories or components). However, it is not specifically optimized for calculating real-time first differences.

2. MOSFiT s another Python tool developed to investigate the secular variation (SV) of the Earth's geomagnetic field. It is useful for detecting geomagnetic jerks and assessing the quality of geomagnetic observatory data. MOSFiT works with data from any INTERMAGNET geomagnetic observatory that is sampled at one-minute intervals.

In contrast, our tool, also based on a Python library, offers immediate insights into rapid changes in the geomagnetic field between consecutive measurements (e.g., $\Delta B / \Delta t$) on a per-second basis. This tool has been under observation for the past few days to evaluate the performance of recording systems and data quality in real-time. It is designed to be simple and computationally efficient.

Comment-2: It would be helpful to include a flowchart or schematic illustrating the processes involved in the automated QC and data transfer pipeline. This would aid the reader in understanding how the different components interact and when human intervention is required.

Reply: Here's flowchart or schematic illustrating the processes involved in the automated QC and data transfer pipeline

[Figure]

Comment-3: To my understanding, the service is still under development. However, the manuscript should provide a more precise explanation of how the automatic QC process functions in practice. The current description mentions that the QC is based on First Differences (FD), but this approach can mistakenly flag legitimate geomagnetic variation (e.g. during storms) as

noise. Do you automatically discard all FD-flagged data, or are these values manually reviewed? Clarifying this possibly within the flowchart is essential.

Reply: The QC is based on First Differences (FD), but this approach will not automatically discard all the FD-flagged data. The flagged data points between the observatories (CPL and HYB) are manually reviewed and then removed.

Comment-4: It would be beneficial to discuss the risks associated with automated data cleaning, such as inadvertently removing valid data, and how these risks are mitigated (e.g., threshold tuning, post-flagging review, cross-validation with secondary data streams).

Reply: Thank you for raising this important point. Automated geomagnetic data cleaning does carry inherent risks, such as the unintended removal of valid signals or the misclassification of noise. To mitigate these risks, we employ several strategies:

- Threshold Tuning: The parameter for outlier detection for +/-0.2nT are defined by INTERMAGNET.
- Post-Flagging Review: Rather than being removed immediately, suspect data points are flagged. This allows for manual verification when necessary.
- Cross-Validation: Multi-station comparisons of CPL for HYB and HYB for CPL will be used to confirm anomalies and ensure consistency.

Comment-5: Please maintain consistent notation for data cadence (e.g., "1 sec", "1 min") throughout the text.

Reply: Modified the data cadence as suggested throughout the text.

Comment-6: -Are the Python scripts or software packages publicly available? If so, a GitHub or repository link would be appreciated.

Reply: At present, the Python scripts and associated software packages are not publicly available, in adherence to directives from our Director and institutional policies governing data and software dissemination.

However, we are committed to supporting the scientific community and would be glad to explore avenues of collaboration or provide specific assistance to observatories or institutions that may require it. Please feel free to reach out with any specific needs or proposals.

Comment-7: The term "real-time" should be clearly defined. For example, does it mean a latency of less than 5 minutes?

Reply: Yes, in real-time with a latency less than 5 minutes.

Comment-8: The website shown in the plots appears to be inaccessible. If it is not publicly available, please state this explicitly in the manuscript and clarify whether public access is planned for the future.

Reply: The website shown in the manuscript is currently inaccessible, but it is accessible within the institute. We plan to make this website available for public access in the future.

Comment-9: Section 4 (Upgrading the PHP server to a Python server): It is unclear whether the migration to Python Django and Bokeh has already been completed, or if this remains part of future plans. Please clarify the implementation status.

Reply: Yes, migration to Python Django and Bokeh has already been completed and implemented.

Comment-10: L86: No need to redefine the abbreviation GIN, as it is already explained in L57.

Reply: Removed the abbreviation GIN as suggested.

Comment-11: L130: Same comment — GIN is already defined earlier.

Reply: Removed the abbreviation GIN as suggested.

Comment-12: L154: The phrase "weekday's data" is ambiguous. Do you mean data from one day or from an entire week?

Reply: The term "weekday data" refers to storing data from day one to day seven, or collecting data for one week. In the upgrading process, we enhanced the server's storage capacity to several months (L156).

Comment-13: L200–202: The sentence is confusing and should be revised for clarity and word order.

Reply: The term FD refers to the difference between consecutive values in a dataset. A few words have also been removed from the sentence to avoid confusion (L236-237).

Comment-14: L226–227: The statement implies that 1-minute data is noisier than 1-second data. However, Figures 2 and 3 suggest the opposite. Please rephrase or clarify.

Reply: Yes, Figures 2 and 3 suggest that 1s data is noisier than 1 min data. Corrected the text as suggested (L263-264).

Comment-15: L244–246: Presenting a long list of numerical values in the text is difficult to follow. Consider using a table for clarity.

Reply: Included the table for clarity

|  | HYB | D(nT) | H(nT) | Z(nT) | F(nT) |
|---|---|---|---|---|---|
| FD: 1min | CPL | ±0.5 | ± 1.5 | ± 0.3 | ± 1.5 |
|  | HYB | ±0.5 | ± 1.5 | ± 0.3 | ± 1.5 |
| FD: 1s | CPL | ± 0.1 | ± 0.1 | ± 0.5 | ± 0.5 |
|  | HYB | ± 0.1 | ± 0.1 | ± 2 | ± 2 |

Comment-16: L355: It would be helpful to describe how the thresholds used in QC are determined are they fixed, empirical, or adaptively set?

Reply: Included in the text: Lines: 397-410.

Comment-17: L357–359: The sentence suggests that only specific devices can run the tool. Is this a hard requirement, or are these just tested and recommended devices? Please clarify.

Reply: Thank you for your insightful question. The devices listed—such as Raspberry Pi, Omega2 LTE, Libre Computer Board Le Potato, and others—are representative examples of low-power, remote-deployable hardware platforms that align well with the intended application environment. Although the current Python-based tool has not undergone formal testing on these specific devices, its modular and platform-agnostic architecture allows for straightforward extension and adaptation to such hardware. We are confident

that, with necessary environment-specific adjustments and dependency management, the tool can be effectively deployed on these and similar devices to facilitate real-time data acquisition and quality control in remote observatory settings.

Comment-18: A brief performance benchmark (e.g., number of flagged points per day, false positive/negative rates if tested) could support claims about the system's effectiveness.

Reply: An example is already illustrated in Figure 4. Another instance is presented below, where we observed a sudden increase in the amplitude of the vector components at CPL due to the impact of a lightning strike. Our tool recorded this event during the real-time quality control check, and it was not deleted. These natural anomalies will be flagged during the data processing phase. The same event is not observed in HYB.

[Figure]

.